# Exploring Unmet Needs from an Online Metastatic Breast Cancer Support Group: A Qualitative Study

**DOI:** 10.3390/medicina57070693

**Published:** 2021-07-07

**Authors:** Aravinthan Kadravello, Seng-Beng Tan, Gwo-Fuang Ho, Ranjit Kaur, Cheng-Har Yip

**Affiliations:** 1Breast Cancer Welfare Association Malaysia, Petaling Jaya 46200, Malaysia; aravinharrish@gmail.com (A.K.); ranjit.pritam54@gmail.com (R.K.); 2Faculty of Medicine, University Malaya Medical Centre, Kuala Lumpur 59100, Malaysia; pramudita_1@hotmail.com (S.-B.T.); gwofuang@gmail.com (G.-F.H.); 3Breast Centre, Subang Jaya Medical Centre, Subang Jaya 47500, Malaysia

**Keywords:** unmet needs, metastatic breast cancer, low and middle-income countries

## Abstract

*Background and Objective*: Despite the increasing treatment options for patients with metastatic breast cancer (MBC), unmet needs remain common, especially in low and middle-income countries where resources are limited and MBC patients face many challenges. They often join support groups to cope with their unmet needs. Currently, many MBC patients connect with each other via online support group in view of the constant availability of support and rapid information exchange. The objective of this study is to determine the unmet needs of women with MBC from an online support group. *Material and Methods*: Messages in an online support group of twenty-two MBC patients over a period of three years from August 2016 till August 2019 were thematically analyzed. *Results*: Three themes were generated, (1) unmet information needs (2) unmet financial needs (3) unmet support needs. Women needed information on side effects of treatment, new treatment options and availability of clinical trials. Although Malaysia has universal health care coverage, access to treatment remains a major challenge. When treatment was not available in the public hospitals, or waiting lists were too long, women were forced to seek treatment in private hospitals, incurring financial catastrophe. Insufficient private insurance and inadequate social security payments force many women to consider stopping treatment. Women felt that they were not getting support from their clinicians in the public sector, who were quick to stop active treatment and advise palliation. On the other hand, clinicians in the private sector advise expensive treatment beyond the financial capability of the patients. Women with families also face the challenge of managing their family and household in addition to coping with their illness. *Conclusions*: There is a need for healthcare professionals, policy makers, and civil society to better address the needs of MBC patients through patient-centered, multidisciplinary and multi-organizational collaboration.

## 1. Introduction

Metastatic breast cancer (MBC) is a common presentation in low-and middle-income countries (LMICs) due to lack of knowledge, fear and belief in traditional medicine [1]. 10% of Malaysian women with breast cancer were metastatic at diagnosis. Because of late presentation, a significant proportion of non-metastatic women at diagnosis eventually develop metastases, with a 5-year survival of 69% [2]. Women with MBC face many physical, psychological and social challenges, among them, insufficient health care information and inadequate supportive care [3,4,5]. Access to optimal care is limited as healthcare resources are mainly focused on treatment of early breast cancer [6,7].

The majority of breast cancer support groups deal with women with early breast cancer, and are therefore ill-equipped to cater for women with MBC. How then do these women with MBC deal with their diagnosis, symptoms of metastases, side effects of different treatments, and the increasing cost of cancer care? Women with MBC often seek help from online support groups in addition to their clinicians to address their unmet needs during this period of uncertainty [8,9].

Online support groups consist of a community of people with similar experiences, in this case, MBC, to address the lack of peer support pertaining to psychosocial issues faced by patients with MBC. These groups can vary from websites to social networking platforms and are hosted by cancer organizations with skilled moderators to ensure effective and relevant communication [10]. Social networking support groups have risen in popularity as it provides easy access for sharing information, instant feedback and a user-friendly interface [10,11,12].

Compared to conventional interview based studies, analysing online chat groups qualitatively provides the ability to capture patients’ voices which are closer to their vernacular intent, enabling an in-depth understanding of their needs that the healthcare community may not be aware of [13]. By identifying these gaps in care for MBC patients from online support groups, the relevant help can be acquired to better address their unmet needs. Therefore, the aim of this study is to explore the unmet needs shared within the MBC social networking community.

## 2. Material and Methods

Messages from an online MBC WhatsApp support group were thematically analysed to identify their unmet needs. This WhatsApp support group was formed independently by Breast Cancer Welfare Association (BCWA) as an ongoing platform to provide support for MBC patients. The study itself was conducted in September 2019, three years after the group was established. Members consisted of patients undergoing treatment in University Malaya Medical Center (UMMC) who were then recruited by UMMC breast surgeons. The group chat is moderated by a breast cancer survivor who is a trained counselor. Before data extraction, all the group members were informed of the retrospective study via WhatsApp message and encouraged to privately message the authors with concerns or whether they wanted their posts excluded from the analysis. Verbal informed consent was acquired by the moderator and there were no objections from all the group members. Anonymity of the patients and their messages were ensured by assigning a unique numeric identifier for each participant. Ethical approval was granted by the UMMC Ethics Board (MECID No: 20191022-7933) on 27 November 2019.

WhatsApp conversation history since the inception of the group was emailed via text file from the moderator’s WhatsApp account to the principal investigator’s email. Messages from 15 August 2016 to 14 August 2019 were then identified and downloaded into a qualitative analysis software (NVIVO 12) by the principal investigator. Only alphanumeric text-based messages were included, thereby removing posts that contained only emojis, graphics, audio or video. Each message was accompanied with the date and time posted along with the numeric identifier of the respective participant. 

Qualitative thematic analysis was used based on Braun and Clarke to analyse the messages [14]. The principal investigator, a medical doctor, performed the coding of all the messages, supervised by a co-author, a palliative care physician, who is experienced in qualitative methodologies. Felt needs based on Bradshaw’s taxonomy of needs are considered as unmet needs and coded in this study [15]. Felt needs are changes deemed necessary by patients due to deficiencies in their healthcare, community and lifestyle pertaining to their metastatic disease. These codes were then reviewed by all the authors and manually categorized into different themes based on the intent of each code. The precise interpretation of each code was ensured through constant comparison, discussion and agreement by all the authors. The themes were then tested and revised against the codes to ensure they were representative of the messages. 

## 3. Results

Forty women joined the group chat but only twenty-two women conversed in the span of three years. Six out of twenty-two women were diagnosed with de novo MBC, eleven patients developed metastases within five years of breast cancer diagnosis and treatment, five women developed metastases more than five years after the initial diagnosis and treatment. Four out of the twenty-two women passed away due to cancer during the three-year span of the study, while two women left the chat group following the death of fellow members with only sixteen remaining at the end of the study. Table 1 provides the demographic characteristics of these patients. 

During the three-year span of the study, a total of 13,342 text-only messages were posted in the WhatsApp group. The primary language used was English with less than 1% of Malay language being used. About 60% of these messages were identified as unmet needs. The remainder consisted of info sharing on various breast cancer related events (20%), daily greetings (10%) and discussions on group gatherings (10%). Messages identified as unmet needs had an average of two sentences per post and 15–20 words per sentence. A total of 206 codes were generated. These codes were categorized into twelve subthemes. These subthemes were then grouped into three distinct themes. (1) Unmet information needs, (2) Unmet financial needs, and (3) Unmet support needs. (Figure 1). Regarding data saturation, all twelve subthemes were represented within the first 30 out of the 36 months’ study duration. Selected quotes that are identified to be most representative of the sub themes are included in the results (Table 2, Table 3 and Table 4).

### 3.1. Unmet Information Needs

The greatest unmet information need was on the side effects of treatment, with frequent fatigue, insomnia, and numbness being the common side effects experienced, “*Anybody having side effects with Eribulin? It works for me, but having tiredness and leg cramps. Now having bad numbness in fingers and toe.”* (Patient 1). Information on other pain management options apart from opioid-based drugs and radiotherapy were repeatedly sought after, “*I still feel pains/aches every day. I already did radiotherapy. What else can I do? Tramadol and Codeine helps but when taken too long, it made me drowsy and non-functional”* (Patient 12).

Many expressed a need for cheaper alternative treatments and considered complementary therapy like acupuncture to manage cancer pain or side effects of cancer treatments, “*Anyone did acupuncture to manage pain or side effects of cancer? My private doctor is usually against alternative treatment like acupuncture.”* (Patient 10). There was a demand for information about ongoing clinical trials and cancer immunotherapy among MBC patients, “*Can anyone share how to get into clinical trial in private hospitals? Need to change treatment soon cause I can’t afford it”* (Patient 5). *“What is the latest development for breast cancer therapy in the aspect of immunotherapy? Those offered by outside clinics, need to be wary as it’s not proven and very expensive.”* (Patient 18).

Information on social security assistance which is a form of financial aid provided by the national Social Security Organization (SOCSO) [16] were frequently raised throughout the study as many face difficulties obtaining this financial aid due to the complicated application process and high rejection rate, “*Anyone here applied for SOCSO? My application got rejected again. Anyway I can apply?”* (Patient 22). Access to universal healthcare has enabled patients to seek treatment from the private and public sector. However, patients have expressed confusion in their treatment process due to conflicting information from their public and private sector healthcare providers, “*Dr … [private oncologist] said I can take this oral chemo, but oncologist at. [government hospital] said this should be 3rd line treatment for me...don’t know what to do.”* (Patient 9).

### 3.2. Unmet Financial Needs

The high cost of medication led to substantial financial difficulties for patients and their families with many having considered discontinuing their treatment or starting treatment despite being insured and on assistance programs, “*More than one of us (I have thought of it) are thinking of stopping treatment because we don’t want to use our family budget just to keep the cancer fighter alive. The drugs for us are usually on the very high end of the expenditure bracket especially from private hospitals.”* (Patient 4). 

Private hospital patients frequently faced similar issues trying to fund their treatment, resulting in preference for cheaper alternatives. Patients frequently faced difficulties paying for necessary radiological investigations. Due to the long waiting list in public hospitals, those who were more financially capable preferred to do their scans at private hospitals because of shorter waiting times, “*I’ll need to do a CT scan and was wondering if anyone knows any private hospitals offering affordable and quality services? For public hospital, there will be a long wait list, or need to see medical officer first, go another day etc. - too stressful to do all the running around*.” (Patient 14).

Many patients regretted their decision for not buying medical insurance earlier as it could have prevented or delayed their financial problems. Interestingly, those with insurance have expressed insufficient coverage by their insurance policies to pay for their treatments, especially for MBC, “*I got the shock of my life when I found out my medical insurance coverage for critical illness is limited to RM30K per annum regardless for cancer or admission for fever. I think this is discrimination.”* (Patient 11).

Patients expressed the need for more generic and biosimilar drugs to be accessible locally instead of having to fork out huge sums of money to purchase the original versions just to stay alive. However, there were concerns about the efficacy and safety of generic and biosimilar drugs especially from foreign pharmaceuticals, *“I hope cheaper generic drugs are accessible in the future. We need it to prevent monopoly & reduce costs so more of us can afford them. Hopefully, the generics are as good as the original and safe to use. Let’s hope we have all the options available.”* (Patient 5).

### 3.3. Unmet Support Needs

Unmet support needs in this study are defined as the lack of efforts to address the psychological, physical and emotional problems faced by MBC patients. Many sought psychological and emotional support from their clinicians when they encountered unbearable symptoms, about to start a new treatment or when faced with financial constraints. One patient expressed a lack of emotional support from her oncology team as she felt they were reluctant to find further treatment solutions and was bluntly told to forgo her efforts for anti-cancer treatments, *“They are reluctant to treat MBC. I was told to take morphine & prepare to sit in a wheelchair. I was so traumatised that I cried, she (oncologist) was so sarcastic saying she will refer me to counsellor because I’m scared of dying. Gave up on them.”* (Patient 3).

There is high demand for allied healthcare services specially to manage their physical supportive needs, “*Climbing up and down the stairs is definitely a challenge. I strongly recommend physiotherapy … The request for a nutritionist’s advice and physiotherapist should come from you.”* (Patient 8). Patients have repeatedly expressed a lack of psychological and emotional support from their social circle such as family, friends and community, “*I truly understand the fear, the chaotic schedule and at the same time managing the kids and household, not forgetting the politics of managing family members…extremely exhausting.”* (Patient 15), “*Just feeling down as my friends are traveling here & there while I have to skimp to pay for my treatment. My hubby not so supportive so my friends send me to hospital. Sometimes I feel so down, I am now taking Lexapro to help me feel happier. But I do feel grateful for my friends who also helped me, especially cancer survivors.*” (Patient 13).

Many experienced loneliness, isolation and depression due to the lack of family support. Financial issues and side effects of treatment resulted in these patients feeling socially deprived as they were unable to attend social events and travel with their family members. Patients expressed the importance of having cancer peer support groups especially for those who lacked adequate psychosocial support from their clinicians and family members.

## 4. Discussion

Our results show that despite access to universal health care, MBC patients from middle-income countries still experience difficulties accessing supportive care services specifically medical information, treatment and psychosocial support. The Breast Health Global Initiative (BHGI) recommends a stratified approach in LMICs by integration of a supportive care program, encompassing breast cancer care from diagnosis through curative treatment to survivorship, metastatic disease and end-of-life care [17]. 

Although public hospitals provide highly subsidized cancer care, patients still face problems such as lack of manpower and resources, with long waiting times for oncology appointments coupled with lack of access to expensive life-extending treatment for MBC. The Ministry of Health has developed guidelines on the management of breast cancer. However, these guidelines were released in 2010 and emphasize mainly on the treatment of early breast cancer with palliative care being the only recommendation for MBC [18].

Lack of information on symptoms of metastases, treatment options, and side effects of treatment were among the unmet needs shared among MBC communities globally [19,20,21]. This study has identified many of these unmet information needs were due to MBC patients being unsatisfied with the information obtained from their clinicians. This resulted in many of our patients seeking information from online support groups due to its global community, fast information exchange and constant availability beyond regular medical consultation hours. Despite the vast pool of information, there is a need for accurate information within MBC support groups to prevent conflicting information from their clinicians [20].

We found that MBC patients tend to receive conflicting oncologists’ recommendations from the private sector and public sector, not realizing that the difference in opinions is due to the public sector being limited in their choice of drugs for MBC. This further propels patients to seek information from other sources, primarily online support groups. The need for more information can be met by providing patients opportunities to report their symptoms, side effects of treatment, type of treatment, and their psychosocial problems during consultation [22]. Better communication between patients and healthcare professionals is crucial to prevent misinformation among this group of women.

There were similarities with various MBC support groups on unmet information needs such as clinical trials and immunotherapy [20,22,23]. Media reports sensationalizing some miraculous cures from immunotherapy has led to a cautious yet positive outlook towards cancer immunotherapy, mirroring attitudes towards immunotherapy among health care professionals [23]. Clinical trials on MBC are lacking locally, and only available in a few large cancer centers, therefore clinicians should be well informed about available potential clinical trials that may suit their patients. 

Social security has long been a vital source of monetary assistance for medical care throughout the globe. The Social Security Organization (SOCSO) in Malaysia supports women with terminal illnesses such as MBC with a monthly pension through its Invalidity Pension Fund [16]. However, this applies only to women who are working and contributing to social security during the time they developed breast cancer. MBC patients have expressed frustration with the complicated and tedious application process to receive this aid. There is a need for a financial navigator who can guide them through the application process [24].

Another recurrent unmet need shared across the MBC community from middle-and low-income countries is the lack of financial resources for optimal treatment [6,7]. The ACTION (ASEAN Costs in Oncology) study has shown that in South-East Asia, patients with advanced cancer are more likely to experience financial hardship (defined as a household spending more than 30% of the household income on out-of-pocket payments for cancer treatment) [25]. MBC patients from this study face financial hardships from cancer treatment as well as from non-medical payments related to cancer care; such hardships affect their quality of life, mental health and adherence to treatment [24]. With costly drugs (such as the anti-HER2 therapy) being the major source of patients’ financial struggles, policy makers should consider collaboration with the pharmaceutical industry to provide more affordable yet effective assistance programs for MBC. Long waiting times for consultation, diagnostic and imaging services in public hospitals has led to patients opting for services in private hospitals, hence increasing their financial burden [26]. 

Financial hardship seems to be more pronounced when patients are uninsured [27]. Regret about not having medical insurance was discussed many times. Even those with insurance coverage experienced financial problems when their insurance was insufficient to cover the cost of their treatment. Despite universal health coverage, patients still experience limited access to expensive treatment from public hospitals. Many are forced to sacrifice their family savings in order to afford these costly treatments from private hospitals. There is an urgent need for innovative insurance and non-insurance interventions to ensure MBC patients do not end up in financial ruin [27]. 

In our study, patients preferred the cheaper generic and biosimilar cancer drugs however issues such as quality, manufacturing practice and regulation pose limitations to the access and availability of these drugs. This seems to be a common issue among patients across most middle-income and low-income countries due to limited financial resources [27,28]. Enabling access to these cheaper drugs may provide tremendous financial relief but should never be at the cost of risking the health of patients. 

Many women in this study have expressed lack of support and dissatisfaction with their oncologist, particularly in the public sector. The need for second opinion, lack of referral to allied health care services, such as dieticians and physiotherapists, and poor communication were among the unmet needs faced by these women. This issue is unique to the public sector only due to the time and resource limitations experienced by clinicians in public hospitals thereby affecting their ability to provide timely and professional attention [29].

MBC patients felt that the person they depend on the most, their oncologist, was more ready to stop active treatment for palliation rather than trying possible curative options. Many have expressed anger towards their clinicians when they perceive that private hospitals are making money out of their misery, and towards the government hospital whom they perceive as denying them life-saving treatment. These findings indicate a need for improved communication skills among healthcare professionals especially in aspects of active listening and empathy. 

The stress from home, marriage and family all while trying to survive with metastatic cancer can be overwhelming for these women especially in an Asian country where women often bear the responsibility of managing household duties, raising their kids and contributing to the workforce. There is a lack of awareness among healthcare professionals and the public on the out-of-hospital struggles faced by these patients. The public assumption that once the cancer is cured or under treatment means cancer patients are back to normal is still rampant across middle- and low-income countries resulting in many MBC patients to experience unfair working conditions by their employers.

Despite the lack of support from these parties, cancer patients have expressed the benefits of being part of a cancer support group chat evident from this study. These benefits include obtaining information on treatment options, ways to cope with side effects of treatment and emotional support from their fellow sisters. Some patients felt grateful for these chats, especially during difficult periods of their metastatic journey reminding them they are not alone.

### 4.1. Implications in Clinical Practice

These findings have many implications in clinical practice as it identifies the unmet needs specific to MBC patients and provides healthcare professionals and policymakers a headway on how to better address them. There is a need for better access to up-to-date information pertaining to metastatic breast cancer treatment focusing on its side effects, pain management options and suitable clinical trials specifically on immunotherapy. It is crucial to consider the cost factor whenever deciding on treatment options as it plays a significant role in patients’ financial capacity regardless of their insurance status or access to universal health coverage. With financial catastrophe occurring universally among these patients, they require urgent access to financial support services, financial navigators, innovative insurance interventions, and policies that address the supply-side and demand-side pricing of cancer care services [24,27]. 

Psychosocial support is necessary especially from healthcare professionals, family, and friends to ensure MBC patients do not succumb to poor mental health. MBC support groups should be encouraged as it plays a significant role in providing peer support for patients with significant psychosocial issues. Communication skills training for healthcare providers with a focus on active listening and responding with empathy can help address the unsatisfactory supportive care needs the patients feel that they are getting from their healthcare professionals. To ensure effective psychosocial support is available, these women must have access to relevant cancer support groups when needed, and their families need to be better equipped to understand their needs through family education. Extensive collaboration between different stakeholders, health care sectors, pharmaceutical industry and civil society groups such as the Advance Breast Cancer (ABC) Global Alliance established in 2016 by the European School of Oncology is necessary to address the unmet needs of MBC patients [30].

### 4.2. Limitations

The main limitation of this study lies in the coding process whereby only one independent coder analyzed the WhatsApp messages thereby making it impossible to assess inter-rater agreement. To minimize the risk of misinterpretations during coding, the entire process was closely supervised by another author highly experienced in qualitative methodologies. Each code was reviewed and agreed by all the authors to represent unmet needs before being categorized into themes. 

As only 22 patients were involved in this study, these findings are not representative of all MBC patients throughout the country. In the absence of any population data on the demographics of MBC in Malaysia, it is not possible to determine whether this sample is representative in terms of age, ethnicity, education, financial and social background. It is also unusual that there was a lack of discussion about sexuality, end-of-life issues and rarely any needs expressed about palliative care services, although several of the women seem to have problems dealing with pain. We speculate this could be due to death being considered taboo in Asian culture where it is believed that such talk may bring bad luck to the person and hasten her death. There was also no discussion of employment discrimination likely because a majority of participants were no longer working. The group started with twenty-two active patients, with two patients deciding to leave during the study after four of their members passed away at various periods due to cancer in the span of three years. These two patients left the group as they were unable to face the prospect of more of their friends dying. In addition, they left because they found the messages scary especially when their members describe the symptoms of metastases and side effects of treatment.

Further studies focusing on needs of a more diverse group of MBC patients, with those living with metastatic disease for a longer period and how they respond to end-of-life issues should be carried out.

## 5. Conclusions

Women with MBC have unique unmet information, financial and support needs that have been long overlooked due to increased attention on detection and treatment of early breast cancer. These unmet needs may provide health care professionals, policy makers, and civil society a headway on how to better address their needs through patient-centered, multidisciplinary and multi-organizational collaboration.

## Figures and Tables

**Figure 1 medicina-57-00693-f001:**
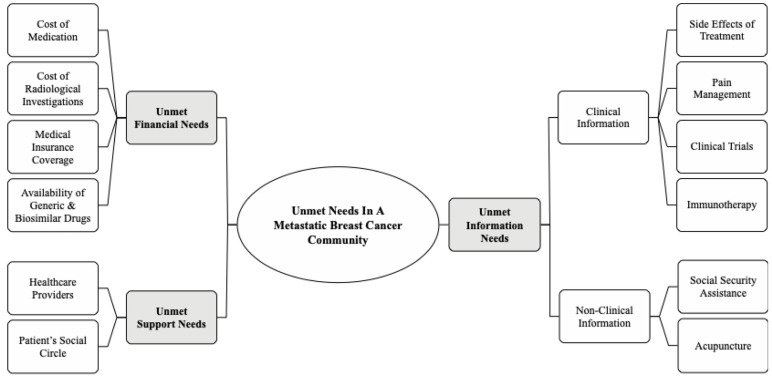
Diagrammatic representation of themes and subthemes.

**Table 1 medicina-57-00693-t001:** Demographic characteristics of participants.

Characteristics		No (%)
Age, Years	41–45	3 (13.6)
	46–50	10 (45.5)
	51–55	5 (22.7)
	56–60	4 (18.2)
Marital Status	Married	19 (86.4)
	Single	3 (13.6)
Ethnicity	Chinese	14 (63.6)
	Malay	5 (22.7)
	Indian	3 (13.6)
Duration of Metastases	<1 year	1 (4.5)
	1–2 years	10 (45.5)
	3–5 years	9 (40.9)
	6–10 years	2 (9.1)
Initial Diagnosis (EBC)	Metastatic	6 (27.3)
	1–5 years	11 (50)
	6–10 years	3 (13.6)
	11–15 years	2 (9.1)
Location of metastases	Bone only	8 (36.4)
	Viscera only	10 (45.5)
	Bone and viscera	4 (18.2)
Chat Status	Remained	16 (72.7)
	Deceased	4 (18.2)
	Left	2 (9.1)

**Table 2 medicina-57-00693-t002:** Unmet Information Needs.

	Unmet Information Needs
Side Effects of Treatment	*Anybody having side effects with Eribulin? It works for me, but having tiredness and leg cramp. Now having bad numbness in fingers and toes. Praying for a miracle.* (Patient 1) *I’m frightened. Terribly frightened. Chemo treatment is suffering is it? I heard patients died during chemo. I’m terrified of nausea, and not having hair. In my whole life, I haven’t been sick, suddenly diagnosed with cancer. I’m scared and lonely.* (Patient 16)
Pain Management	*I am on Xgeva, calcium and vitamin D3 but I still feel pains/aches every day. I already did radiotherapy. What else can I do? Tramadol and Codeine helps but when taken too long, it made drowsy and non-functional.* (Patient 12)
Clinical Trials	*Can anyone share how to get into clinical trial in private hospitals? Need to change treatment soon cause I can’t afford it.* (Patient 5)
Immunotherapy	*What is the latest development for breast cancer therapy in the aspect of immunotherapy? Those offered by outside clinics, need to be wary as it’s not proven and very expensive.* (Patient 18)
Acupuncture	*Anyone did acupuncture to manage pain or side effects of cancer? My private doctor is usually against alternative treatment like acupuncture.* (Patient 10)
Social Security Assistance	*Anyone here applied for SOCSO? My application got rejected again. Any way I can apply? Most of us have had careers and we have contributed towards the economy.* (Patient 22)
Types of Treatment	*Dr … (private oncologist) said I can take this oral chemo, but oncologist at … (government hospital) said this should be 3rd line treatment for me...don’t know what to do. (Patient 9)*

**Table 3 medicina-57-00693-t003:** Unmet Financial Needs.

	Unmet Financial Needs
High Cost of Treatment	*More than one of us (I have thought of it) are thinking of stopping treatment because we don’t want to use our family budget just to keep the cancer fighter alive. The drugs for us are usually on the very high end of the expenditure bracket especially from private hospitals*. (Patient 4)
Cost of Radiological Investigations	*I’ll need to do a CT scan and was wondering if anyone knows any private hospitals offering affordable and quality services? For public hospital, there will be a long wait list, or need to see medical officer first, go another day* etc. *- too stressful to do all the running around.* (Patient 14)
Medical Insurance Coverage	*I got the shock of my life when I found out my medical insurance coverage for critical illness is limited to RM30K per annum regardless for cancer or admission for fever. I think this is discrimination.* (Patient 11)
Availability of Generic & Biosimilar Drugs	*I hope cheaper generic drugs are accessible in the future. We need it to prevent monopoly & reduce costs so more of us can afford them. Hopefully, the generics are as good as the original and safe to use. Let’s hope we have all the options available.* (Patient 5)

**Table 4 medicina-57-00693-t004:** Unmet Support Needs.

	Unmet Support Needs
Healthcare Providers	*Don’t waste time checking with public hospitals. They are reluctant to treat MBC. I was told to take morphine & prepare to sit in a wheelchair. I was so traumatised that I cried, she (oncologist) was so sarcastic saying she will refer me to counselor because I’m scared of dying. Gave up on them.* (Patient 3) *Climbing up and down the stairs is definitely a challenge. I strongly recommend physiotherapy. Don’t wait for your oncologist to do it. The request for a nutritionist’s advice and physiotherapist should come from you. With our disease, we really need to be brave and upfront to some extent. Be your own advocate!* (Patient 8)
Patient’s Family And Social Circle	*Just feeling down as my friends are traveling here & there while I have to skimp to pay for my treatment. My hubby not so supportive so my friends send me to hospital. Sometimes I feel so down, I am now taking Lexapro to help me feel happier. But I do feel grateful for my friends who also helped me, especially cancer survivors.* (Patient 13) *I truly understand the fear, the chaotic schedule and at the same time managing the kids and household, not forgetting the politics of managing family members......extremely exhausting* (Patient 15)

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
