# Peer review of "Exploring Unmet Needs from an Online Metastatic Breast Cancer Support Group: A Qualitative Study"

_medicina, 2021, doi:10.3390/medicina57070693_

Round 1
Reviewer 1 Report
Kadravello et al., 2021 in their qualitative study analyzed the kind of unmet needs discussed by patients in a Whatsup group. The data is derived from a limited number of patients (16 active patients). The outcome of the study is not surprising, however, is important. It would be better and more relevant if the data were derived from more subjects (more than 50). However financial constraints and journal in which authors are submitting, this patient's number may be enough. I leave this decision to the Editor's end.
The study is well within the scope of the journal however authors need to address several points before being considered.
- Abstract: Please specify the duration of the study (three years: month and year).
- Since it is a retrospective study: why patients chat was retained for a period of three years. Since this study is being considered for publication in the year 2021, why only three years were considered. Is this WhatsApp group is no longer active? Please provide these information in materials and method section.
- Is written consent from patients was taken? If yes, please include a sentence in the materials and methods section.
- In discussion: Please specify if patients get benefitted from chats among themselves. If yes please provide some more information. Such information is important for the future perspective of this study.
- Line 340-341: Is the reasons were provided by patients or it is just speculation. Please specify.
- What was the language used in the chat should be included in table Table 1.
- Line 103; 60% of these messages were identified as unmet needs. What are the criteria and keywords that authors used to identify unmet needs? Please include these criteria.
- Finding: The name of the private donation agency should be included.
- Table 2: Do authors observed any religious faith and cancer survival rate in unmet chat?
Minor
Line 35: [1] should be within full stop (.).
Author Response
Response to Reviewer 1
Thank you for your comments. We hope we have addressed them to your satisfaction.
Point 1: Abstract: Please specify the duration of the study (three years: month and year).
We have included the details as suggested.
Messages in an online support group of twenty-two MBC patients over a period of three years from August 2016 till August 2019 were thematically analyzed.
Point 2: Since it is a retrospective study: why patients chat was retained for a period of three years. Since this study is being considered for publication in the year 2021, why only three years were considered. Is this WhatsApp group is no longer active? Please provide these information in materials and method section.
We hope our following response will be able to address them accordingly.
Changes in Materials & Methods Section:
This WhatsApp support group was formed independently by Breast Cancer Welfare Association as an ongoing platform to provide support for MBC patients. The study itself was conducted in September 2019, three years after the group was established.
Point 3: Is written consent from patients was taken? If yes, please include a sentence in the materials and methods section.
Thank you for the comments. Most patients in this group chat were located in different regions of the country. Considering the logistical challenges involved in obtaining written consent from all of them, all the patients opted for informed verbal consent and it was obtained by the group moderator through personal messaging.
Changes in the Materials & Methods Section:
Before data extraction, all the group members were informed of the retrospective study via WhatsApp message and encouraged to privately message the authors with concerns on whether they wanted their posts excluded from the analysis. Verbal informed consent was acquired by the moderator and there were no objections from all the group members.
Point 4: In discussion: Please specify if patients get benefitted from chats among themselves. If yes please provide some more information. Such information is important for the future perspective of this study.
Thank you. Throughout the span of three years, patients were able to exchange information on treatment, resources for cancer care and ways to cope with their illness. Patients have expressed the usefulness of group chats in providing emotional support throughout their metastatic journey.
Changes to Discussion:
Despite the lack of support from these parties, cancer patients have expressed the benefits of being part of a cancer support group chat evident from this study. These benefits include obtaining information about treatment options, ways to cope with side effects of treatment and emotional support from their fellow sisters. Some patients felt grateful for these chats, especially during difficult periods of their metastatic journey reminding them they aren't alone.
Point 5: Line 340-341: Is the reasons were provided by patients or it is just speculation. Please specify.
We have included more details as suggested.
Changes to Discussion:
It is also unusual that there was a lack of discussion about sexuality, end-of-life issues and rarely any needs expressed about palliative care services, although several of the women seem to have problems dealing with pain. We speculate this could be due to death being considered taboo in Asian culture where it is believed that such talk may bring bad luck and hasten death.
Point 6: What was the language used in the chat should be included in table Table 1.
We have included further details as suggested.. The primary language used was English with less than 1% of Malay language used mainly to describe certain words. We have clarified this in the results section.
Changes to Results:
The primary language used was English with less than 1% of Malay language used.
Point 7: Line 103; 60% of these messages were identified as unmet needs. What are the criteria and keywords that authors used to identify unmet needs? Please include these criteria.
We have included more details as suggested.
Changes to Materials & Methods:
Felt needs based on Bradshaw’s taxonomy of needs are considered as unmet needs and coded in this study. [15] Felt needs are changes deemed necessary by patients due to deficiencies in their healthcare, community and lifestyle pertaining to their metastatic disease.
Point 8: Finding: The name of the private donation agency should be included.
We have included the relevant details as suggested in the section on Funding.
This research was funded by an anonymous donor with the funds being used to pay for a research assistant.
Point 9: Table 2: Do authors observed any religious faith and cancer survival rate in unmet chat?
Response 9 : We appreciate your comments and hope to address them in the following sentence. We did not explore patients' religious faith and its relation to cancer survival rate in this study. We have not come across any patients expressing a need for religious intervention in the chat group during the span of three years. It was also not possible to identify the cancer survival rate as patients were not keen on sharing this information with us.
Point 10: Minor Line 35: [1] should be within full stop (.).
Response 10 : Thank you once again for your comment. We have made the changes as recommended.
Changes to Introduction:
Metastatic breast cancer (MBC) is a common presentation in low- and middle-income countries (LMICs) due to lack of knowledge, fear and belief in traditional medicine [1].
Reviewer 2 Report
In this manuscript, the authors classified the unmet needs of women with metastatic breast cancer by analyzing messages in an online support group. The messages were collected over a period of three years with 22 MBC patients participating. The authors’ approach of research is novel, and the result drawn from the message analysis reflected important current unmet information/financial/support needs that the whole society could pay attention to and manage to solve.
Typo:
Page 2, line 66: the abbreviation of University Malaya Medical Centre (UMMC) should be introduced when it first appears.
Page 6, line 183-185, 187-193: use italics for the words of patients.
The format of ref.
Page 1, line 40: additional period after the ref
Page 1, line 41: additional period after the ref
Page 2, line 58: missing period before the ref
Page 7, line 212: additional period after the ref
Page 7, line 244: additional period after the ref
Page 8, line 270: repeated ref
Page 8, line 278: additional period after the ref
Page 9, line 318: additional period after the ref
Overall, this manuscript is worth publishing.
Author Response
Response to Reviewer 2
Thank you very much for your encouraging comments
Point 1: Typo:
Page 2, line 66: the abbreviation of University Malaya Medical Centre (UMMC) should be introduced when it first appears.
Page 6, line 183-185, 187-193: use italics for the words of patients.
Thank you once again for your recommendations. We have included the amendments as suggested.
Point 2: The format of ref.
Page 1, line 40: additional period after the ref
Page 1, line 41: additional period after the ref
Page 2, line 58: missing period before the ref
Page 7, line 212: additional period after the ref
Page 7, line 244: additional period after the ref
Page 8, line 270: repeated ref
Page 8, line 278: additional period after the ref
Page 9, line 318: additional period after the ref
We apologize for the minor mistakes. We have addressed them as suggested to improve the manuscript.
Overall, this manuscript is worth publishing.
Thank you so much!
Round 2
Reviewer 1 Report
Yip and the team have made good efforts in improving their manuscript that reflect in the revised version. The manuscript is now ready for publication after minor typographical corrections. The present study is a good example of how scientific studies can be conducted effectively using the least resources. The manuscript is within the scope of the journal and may be accepted for publication.
My heartiest congratulations to all the authors.